# Move Well, Feel Good: Feasibility and acceptability of a school-based motor competence intervention to promote positive mental health

**Stuart J. Fairclough**[1]*, **Lauren Clifford**[1], **Lawrence Foweather**[2], **Zoe R. Knowles**[2], **Lynne M. Boddy**[2], **Emma Ashworth**[3], **Richard Tyler**[1]

1 Department of Sport and Physical Activity, Sport, Physical Activity, Health and Wellbeing Research Group, Edge Hill University, Ormskirk, Lancashire, United Kingdom, 2 The Physical Activity Exchange, Research Institute for Sport and Exercise Sciences, Liverpool John Moores University, Liverpool, United Kingdom, 3 School of Psychology, Liverpool John Moores University, Liverpool, United Kingdom

* stuart.fairclough@edgehill.ac.uk

**Data Availability Statement:** The data are available from https://osf.io/qfmc7/.

**Funding:** This work was supported by a grant from The Waterloo Foundation (#1669/4195) that was

## Abstract

### Background

In response to the adverse impacts of the COVID-19 lockdown measures Move Well, Feel Good (MWFG) was developed as a school intervention using improvement of motor competence as a mechanism for promoting positive mental health. Study objectives were to evaluate the feasibility and acceptability of MWFG and to describe changes in child-level outcomes.

### Methods

Five northwest England primary schools were recruited. MWFG was delivered over 10-weeks through physical education (PE) lessons, which were supplemented by optional class-time, break-time, and home activities. The intervention focused on development of 9–10 year-old children's motor competence in locomotor, object control, and stability skills, and psychosocial skills. Feasibility was evaluated against nine pre-defined criteria using surveys, interviews (teachers), and focus groups (children). Pre- and post-intervention assessments of motor competence, mental health, prosocial behaviour, wellbeing, and 24-hour movement behaviours were also completed.

### Results

The five recruited schools represented 83% of the target number, 108 children consented (54% of target) with teachers recruited in all schools (100% of target). Intervention dose was reflected by 76% of the 45 scheduled PE lessons being delivered, and adherence was strong (>85% of children attending ≥75% of lessons). Positive indicators of acceptability were provided by 86% of children, 83% of PE teachers, and 90% of class teachers. Data collection methods were deemed acceptable by 91% of children and 80% of class teachers,

awarded to SJF, RT, LF, LMB, ZK, and EA. The funders had no role in study design, data collection and analysis, decision to publish, or preparation of the manuscript.

**Competing interests:** The authors have declared that no competing interests exist.

and children spoke positively about participating in the data collection. Child-level outcome data collection was completed by 65%-97% of children, with a 3%-35% attrition rate at post-intervention, depending on measure. Favourable changes in motor competence (+13.7%), mental health difficulties (-8.8%), and prosocial behaviour (+7.6%) were observed.

## Conclusions

MWFG is an acceptable and feasible motor competence intervention to promote positive mental health. Content and delivery modifications could inform progression to a pilot trial with a more robust design.

## Introduction

The COVID-19 pandemic and associated social distancing measures resulted in unprecedented, enforced changes to UK citizens' routines and lifestyles. Children were particularly affected through school closures, online home learning, the ceasing of organised sports, and restrictions on face-to-face social interactions. The lockdown restrictions were negatively associated with children's mental health and wellbeing [1–3], with those from lower socioeconomic status families disproportionately affected [3, 4]. Furthermore, the restrictions contributed to increases in children's digital screen use [5] and decreases in physical activity [6], particularly structured activities (e.g., physical education lessons, organised leisure time sport participation [7, 8]). These activities are critical for development of motor competence [9] (i.e., the degree to which a child performs goal-directed movements in a coordinated, accurate, and relatively error-free manner [10]), which is an essential foundation for future movement and physical activity behaviours [11]. Given the established relationships between motor competence and physical activity [11], it is unsurprising that reduced activity during the pandemic was associated with attenuated motor competence once lockdown restrictions were lifted [12]. Moreover, poor motor competence is associated with reduced mental health, including internalising difficulties such as anxiety and depression [13], and these associations may be mediated by limited social support mechanisms [13, 14] and low self-perceptions [15, 16]. Improving children's motor competence may therefore be a mechanism for promoting children's mental health, through enhancing aspects of psychosocial development.

The inter-relationships between children's motor competence, mental health, and their mediators are described in the Elaborated Environmental Stress Hypothesis model (EESH) [17]. Empirical support exists for the EESH [18], which posits that poor motor competence predisposes children to internalising mental health difficulties (e.g., anxiety, depression) via interactions with environmental stressors such as low self-esteem, low social support, physical inactivity, being overweight, etc. These stressors can, in turn, be 'buffered' by social and personal resources such as peer and parental support and enhanced perceived competence [18]. Previous studies in community samples have reported favourable relationships between motor competence, internalising difficulties and other psychosocial outcomes described in the EESH model [14, 19].

There is a need for intervention strategies in primary school children to address the well-established low levels of motor competence and poor mental health and wellbeing which declined further as immediate and persisting consequences of the COVID-19 lockdown measures [1, 12], even after they ended [6, 20, 21]. Schools are suitable settings for the promotion of child health and wellbeing [22]. Furthermore, primary school motor competence

interventions can be efficacious for improving motor skills [23, 24] and there is some albeit limited evidence that they can also enhance mental health and wellbeing [25, 26]. However, no such intervention studies involving children without movement difficulties (e.g., developmental coordination disorder) within mainstream schools have been undertaken in the UK. In response to the adverse health and wellbeing consequences of the lockdown restrictions observed in schools, we applied the EESH model to underpin a school-based motor competence intervention to enhance children's mental health entitled 'Move Well, Feel Good' (MWFG).

This study focused on the feasibility of the MWFG intervention. Feasibility studies are an integral part of the Medical Research Council's complex intervention development and evaluation framework, which assess pre-defined progression criteria related to interventions and/or their evaluation [27]. Feasibility studies generate sufficient evidence to make informed decisions about whether an intervention has promise and its findings can be replicated in a scaled-up trial [28]. As the development of MWFG was consistent with the Medical Research Council framework, the primary aim of this study was to evaluate the feasibility and acceptability of MWFG to assess its potential for implementation as a pilot trial. A secondary aim was to describe changes in child-level outcomes that reflected elements of the EESH model.

## Materials and methods

### Study design

This feasibility study was conducted as a single-arm pre-post intervention [29] whereby all children received the same intervention. Pre-intervention (T0) and post-intervention (T1) measures were obtained over two consecutive days one week before (week of 12th September 2022) and after the intervention (week of 12th December 2022), respectively. Additional feasibility data were collected from teachers during each week of the intervention between T0 and T1.

### Sample size estimation for feasibility outcomes

Using pilot study progression criteria Red/Stop upper limit and Green/Go lower limit reference tables specifically develop for feasibility outcomes [30] we estimated a sample size for child recruitment. Based on an alpha level of 0.05 and 90% power to reject being in the Red zone if the Green zone holds true, the minimum required sample size was $n = 46$. Depending on the number of Year 5 classes in each school we proposed to recruit between 100 and 200 children to the study. This is significantly more than the estimate from the power calculation, and consistent with the upper end of the sample size scale ($n = 100$) typically observed in health behavioural pilot and feasibility studies [28].

### Participants and recruitment

Phase 1 of MWFG involved school stakeholders (e.g., children, PE teachers, class teachers, school leaders) and the researchers co-creating the intervention programme [31] with schools and children recruited to the study during this phase between January 31st and March 1st 2022. Briefly, participants were Year 5 children (age 9–10 years) recruited from primary schools in West Lancashire, northwest England. Eligible schools had to be located in low socioeconomic status (SES) areas based on school postcode-linked English Indices of Multiple Deprivation (EIMD; deciles 1–3) [32] and have >18% (England average) of children eligible for free-school meals. Schools meeting these criteria and with at least 25 children per class were contacted to ascertain their interest in the study. All Year 5 children who were physically able to participate

in physical education (PE) lessons were eligible to participate. Written informed consent was obtained for all child (parental/carer consent and child assent) and teacher participants. Ethical approval was received from the Science Research Ethics Committee at Edge Hill University (#ETH2122-0062). The study was registered with the International Standard Randomised Controlled Trial Number (ISRCTN) Registry (#ISRCTN23960783). The study was registered retrospectively and before enrolment of participants started because the design and content of the intervention was unknown at the time of study commencement (i.e., Phase 1 intervention co-creation). The authors confirm that all ongoing and related trials for this intervention are registered.

## MWFG intervention

The core element of MWFG was PE lessons focused on improving motor competence. The lessons were taught in three blocks of three weeks (minimum dose of nine lessons in total), focusing on combinations of locomotor, object-control, and stability skills, respectively. The lessons were developed by the research team and included adaptations of existing physical literacy and PE resources [33, 34]. Specific psychosocial concepts (e.g., perceived competence, self-worth, resilience, co-operative working) were integrated into the PE lesson plans to reinforce these important EESH 'buffers' against risks to positive mental health (see S1 File). The design and content of the lessons were based on established pedagogical principles. Specifically, the SAAFE Framework emphasised supportive, active, autonomous, fair, and enjoyable lessons [35], while assessment for learning [36] was integrated through use of task-oriented resource cards for self- and peer-assessment. PE teachers received a weekly summary lesson plan which cross-referenced activity resource cards that could be used in each lesson. Each lesson also included an associated activity card for children to try at home with family members or during recess with friends (see S2 File). These self-directed individual or group 'skill snacks' were of short duration and related directly to the content of that week's PE lesson. These activity cards and accompanying demonstration videos could be accessed via a QR code on the PE teacher's lesson plans and an electronic link which was sent to each child's parent/carer via the schools' communication systems.

Two weeks before the intervention started the PE teachers delivering the intervention received an online familiarisation session. The online format was used for convenience to accommodate PE teachers' availability and a recording of the session with transcript was shared for attendees and non-attendees to refer to in their own time. PE teachers were encouraged to ask questions or make suggestions to the researchers about the programme subsequently via email, text, or phone. To reinforce PE lesson content, class teachers received suggestions for integration of motor competence activities focused on the same movement skills into their teaching for class-time physically active learning, which also emphasised the importance of incorporating the psychosocial concepts into lessons. All PE lessons, skill snacks resources, and class teacher resources were provided to participating teachers in electronic and hard copy formats.

Following an introductory classroom launch lesson, the core element of MWFG was delivered by PE teachers during one weekly ~45-minute PE lesson between September and December 2022. Schools were closed, as was typical, for one week's break mid-way through this period, during which children were provided with activity cards as 'homework' and encouraged to try them at home. Throughout the intervention, additional digital and video resources for motor competence and psychosocial development were provided electronically to class teachers and parents/carers for use in lessons taught in classrooms or elsewhere in the school grounds, at break-time, and at home.

**Table 1. Intervention feasibility outcome measures summary.**

| Feasibility outcome measures | When administered/ completed | Participants | Feasibility outcomes | Response modes |
|---|---|---|---|---|
| Delivery log | Weekly | PE teachers Classroom teachers | Dose Adherence | Closed question responses |
| Intervention and research methods acceptability surveys | T1 | PE teachers Classroom teachers | Acceptability of intervention Acceptability of data collection methods | Likert scale (Strongly disagree to Strongly agree) |
| Intervention and research methods acceptability survey | T1 | Children | | Likert scale smiley face emoticons (Strongly disagree to Strongly agree) |
| Semi-structured interviews | T1 | PE teachers Classroom teachers | Acceptability of intervention Acceptability of data collection methods | Oral responses to open-end interviewer questions and prompts |
| Participatory focus groups | T1 | 6 children per school | Acceptability of intervention Acceptability of data collection methods | Drawing task and methods photographs to stimulate discussion; Oral responses to open-end interviewer questions and prompts |
| Consent forms | Before T0 | Children PE teachers Classroom teachers | Child and teacher recruitment | Signed informed consent |
| Child-level outcomes | T0 T1 | Children | Data collection and data attrition rates | As per administration protocols for each measure (see Child-level outcomes section) |

## Feasibility outcomes

The uncertainties relating to the feasibility of MWFG and the subsequent progression to a pilot intervention trial represented the primary study outcomes. These related to eligibility and recruitment, intervention implementation (e.g., resources, delivery), intervention adherence and engagement, acceptability of data collection procedures, and data attrition [29]. These feasibility outcomes were measured using quantitative and qualitative methods, which are summarised in Table 1 and explained in detail in S3 File. Nine *a priori* feasibility criteria were agreed with the study funder as progression criteria to a full trial. These were incorporated into a traffic light system (i.e., green: continue to trial, amber: further discussion and changes needed, red: do not proceed to trial) [29, 37] and are detailed in Table 2.

## Child-level outcomes (collected in schools)

**Motor competence.** Motor competence was assessed using the Canadian Agility and Movement Skill Assessment (CAMSA) [38] and three stability skills from the Dragon Challenge [39]. Both the CAMSA and Dragon Challenge were developed and validated to assess motor competence in primary school-aged children. The CAMSA involved children completing a sequence of seven movement skills in a timed agility course. These skills include jumping on two feet through three hoops, sliding sideways between cones, catching, overhand throw, skipping between cones, hopping on one foot through six hoops, and kicking a ball. The quality of the performance of each skill was scored based on 14 movement criteria, each receiving one point if the criterion was present and zero points if absent. The time taken to complete the agility course was recorded, with faster times scoring higher (out of 14). Children had two practice attempts before completing two timed and scored trials. Scores from both trials were summed to provide a CAMSA overall score (range 2 to 56) [38].

**Table 2. Traffic light feasibility progression criteria.**

| Progression criteria | Red (stop) | Amber (discuss and amend) | Green (go) |
|---|---|---|---|
| School recruitment (targeting n = 6) | ≤50% of target number | 50–90% of target number | ≥90% of target number |
| Child participant recruitment (targeting n = 200) | <20% of eligible children | 20–74% of eligible children | ≥75% of eligible children |
| Deliverer recruitment | ≥1 class teacher and ≥1 PE deliverer in <50% of schools | ≥1 class teacher and ≥1 PE deliverer in ≥50%-75% of schools | ≥1 class teacher and ≥1 PE deliverer per school |
| Intervention dose | <40% of scheduled sessions delivered/week | 40–79% of scheduled sessions delivered/week | ≥80% of scheduled sessions delivered/week |
| Intervention adherence | <40% of recruited children attend ≥75% of MWFG PE lessons | 40–69% of recruited children attend ≥75% of MWFG PE lessons | ≥70% of recruited children attend ≥75% of MWFG PE lessons |
| Acceptability of intervention | <50% of teachers and children found MWFG acceptable | 50–79% of teachers and children found MWFG acceptable | ≥80% of teachers and children found MWFG acceptable |
| Acceptability of data collection methods | <50% of teachers and children found data collection methods acceptable | 50–79% of teachers and children found data collection methods acceptable | ≥80% of teachers and children found data collection methods acceptable |
| Secondary outcome data collected at baseline | Data collected from <50% of children | Data collected from 50–74% of children | Data collected from ≥75% of children |
| Follow-up secondary outcome data attrition | >40% data attrition at T1 | 26–40% data attrition at T1 | ≤25% data attrition at T1 |

The three Dragon Challenge stability skills were the balance bench, core agility, and wobble spot [39]. Children observed a demonstration of each skill and had one practice attempt before completing a scored trial. The quality and outcome of each skill were assessed using two technical/process criteria and one outcome/product criterion. One point was given for successful demonstration of each of the technical/process criteria successfully and two points were awarded for each outcome/product criterion. A maximum of four points were awarded for each skill and these were summed to provide a Dragon Challenge stability skills score (range 0 to 12) [39]. The CAMSA and Dragon Challenge stability skills were assessed and coded by trained assessors who had experience in administering and analysing both assessments and had no prior knowledge of the participants' movement skill abilities.

**Mental health.** Internalising and externalising mental health difficulties were measured in classrooms using the Strengths and Difficulties Questionnaire (SDQ) [40] and the Me and My Feelings questionnaire (MMF) [41]. The 25-item SDQ includes five subscales related to perceived emotional difficulties, behavioural difficulties, hyperactivity, peer relationship difficulties, and prosocial behaviour in the last six months. Each subscale consists of five items which are scored on a 3-point scale ranging from 0 ('not true') to 2 ('certainly true'). Items 7, 11, 14, 21, and 25 are reversed scored. Scores for each subscale are computed by summing their respective items and range from 0 to 10. A total difficulties score reflective of overall mental health can also be computed by summing the four mental health difficulties subscales, ranging from 0 to 40, with higher scores reflecting increased mental health difficulties. Prosocial behaviour is not included within the total difficulties score as it is the positive mental health subscale on the SDQ. When the SDQ is used with community samples it is recommended that the broad constructs of internalising difficulties (emotional difficulties and peer relationships), externalising difficulties (behavioural difficulties and hyperactivity), and overall mental health and prosocial behaviour are reported [42]. Computed scores for each of the five subscales and the total difficulties scale can be classified on a four-band categorisation: close to average, slightly raised, high, and very high [43]. Internal consistency of the SDQ for overall mental health, internalising difficulties, externalising difficulties, and prosocial behaviour was

Cronbach's α = 0.73, 0.76, 0.76, 0.72 (T0), respectively. At T1 Cronbach's α values were 0.78, 0.78, 0.77, 0.69, respectively.

The MMF questionnaire is a 16-item school-based measure of child mental health. It covers the domains of emotional difficulties (10-items) and behavioural difficulties (6-items). The children respond to each statement asking how they feel by selecting 'never', 'sometimes', or 'always'. These responses are scored on a 3-point scale ranging from 0 (never) to 2 (always), with item-15 reversed-scored. Scores for overall mental health as well as emotional difficulties and behavioural difficulties are computed by summing their respective items. These range from 0 to 32, 20, and 12, respectively. MMF has demonstrated acceptable validity and reliability [41, 44] and scores for emotional and behavioural difficulties subscales can be categorised as indicating borderline or clinically significant difficulties [41]. Cronbach's α internal consistency values for overall mental health, emotional difficulties, and behavioural difficulties at T0 were 0.84, 0.85, and 0.81, respectively. The corresponding values at T1 were 0.89, 0.85, and 0.82, respectively.

**Wellbeing.** The KIDSCREEN-10 questionnaire was used as a measure of wellbeing. KIDSCREEN-10 is a 10-item questionnaire, which asks participants how they felt in the last week [45]. The questions reflect the factors of physical well-being, psychological well-being, autonomy, parent relations, peers and social support, and school environment, which are derived from the 27-item version of KIDSCREEN. Responses to each question are recorded using 1–5 Likert scale ranging from 1 (Not at all/Never) to 5 (Extremely/Always). Questions 3 and 4 are reverse-scored then scores for each question are summed, before being converted to T-scores using the methodology described in the KIDSCREEN administration manual [46]. Cronbach's α at T0 and T1 were 0.75 and 0.80, respectively.

**Global self-worth.** The Global Self-Worth (GSW) sub-scale of the Self-Perception Profile for Children [47] was used to measure GSW. The GSW sub-scale consisted of 6-items presented in a structured alternative format. This uses statement types such as 'Some children are very happy being the way they are' BUT 'Other children wish they were different'. Children first decided which kind of child they were most like, after which they decided whether the description was 'really true for me' or 'sort of true for me'. Responses were scored on a four-point scale, where 1 indicated the lowest GSW and 4 reflected the highest level of GSW. On the GSW sub-scale, items 1, 2, and 6 were reverse-scored. Internal consistency was Cronbach's α = 0.81 (T0) and 0.84 (T1).

**Peer support.** The peer support sub-scale of the Student Resilience Survey [48] consisted of 12 questions asking about support from other children at school. The children's responses were scored on a 5-point scale ranging from 1 (never) to 5 (always). The mean of the 12 scores provided a measure of peer support [48]. Internal consistency for T0 and T1 was Cronbach's α = 0.90, 0.91, respectively.

**Social influences on physical activity.** Social influences on the children's physical activity were measured using the Social Influences Scale [49]. This consisted of eight statements relating to friends' and family members' influences on the child's activity in the previous two weeks. Each statement was accompanied with a Yes/No response option which was scored as 1 or 2, respectively. Cronbach's α at T0 and T1 were 0.76 and 0.79, respectively.

**Encouraging friends to be active.** A single question was included asking during a typical week how often the children encouraged friends to be active [50]. The response options of 'never', 'sometimes', and 'every day' were scored with 0, 1, and 2, respectively. All questionnaires were completed at T0 and T1 in classrooms with guidance and instructions from the research team and in the presence of the class teachers.

**School-day physical activity and sedentary time.** Physical activity and sedentary time during school were assessed using ActiGraph GT9X triaxial accelerometers. Participants were

asked to wear the devices on their non-dominant wrist for 24-hours per day over 9 consecutive days at T0 and T1. The devices were initialised to record at 100 Hz and the subsequent data were downloaded using ActiLife (versions 6.13.4). The raw data files (gt3x format) were processed in R using package GGIR version 2.8–2 [51]. Signal processing included autocalibration using local gravity as a reference [52], detection of implausible values, and identification of non-wear. Non-wear was imputed by default in GGIR whereby invalid data were imputed by the average at similar time points on other days of the week [53]. School start and end times were used to calculate school day durations and subsequent data processing was relative to these time windows. The raw triaxial accelerometer signals were converted to one summary measure of acceleration (Euclidean Norm Minus-One; ENMO) expressed in milligravitational units (mg) [53]. ENMO values were reduced to 5-s epochs and averaged per school day to represent average acceleration as a measure of activity volume [54]. The intensity gradient represents the inverse relationships between time and activity intensity, and was also calculated as a measure of the children's school day intensity profile [54]. Child-specific non-dominant wrist ENMO cut-points of 48 m*g* [55], 201 m*g*, and 707 m*g* [56] defined estimated sedentary time and light physical activity (LPA), moderate physical activity (MPA), and vigorous physical activity (VPA), respectively. Participants' accelerometer data were included in the analytical sample if valid wear was recorded on at least three valid school days, and if post-calibration error was <10 mg. A valid day was defined as accelerometer wear for at least 95% of the school day. The 5% buffer accounted for short, accumulated periods of recorded non-wear due to sustained periods of stillness during class time. During the accelerometer wear protocol daily rainfall and ambient temperature were recorded for the locations corresponding to each school's postcode (https://www.metoffice.gov.uk).

### Additional measures

**Anthropometrics.** All consenting children had their height and weight measured at T0 and T1 (to the nearest cm/kg) using a portable stadiometer and digital scale (both Seca, Hamburg Germany). Body mass index (BMI) and BMI z-scores (BMIz) were calculated for each participant [57] and international age- and sex-specific BMI cut-points applied to determine weight status [58]. For all measurements, participants wore light clothing with shoes removed in an area away from the other data collection activities.

**Demographic information.** Participants' dates of birth, home postcodes, and ethnicity were obtained from the schools' information management systems. Decimal age and 2019 EIMD scores [32] were calculated using data collection dates and home post codes, respectively. EIMD scores provide an area-level relative measure of deprivation based on income, employment, education, health, crime, housing, and living environment. Area-level SES was represented by the EIMD decile score for each participant. The Family Affluence Scale (FAS) [59] was included in the questionnaire pack that was completed by the children in school. The FAS consists of six questions asking about home and family indicators of affluence (e.g., car ownership, own bedroom, foreign holidays). The response options differed depending on the questions asked but had two, three, or four responses available which were scored from 1 to 2, 3, or 4, respectively. Question scores were summed to give the overall FAS score.

### Recognition of school and child participation

Each school received a bag of multi-skills and physical activity equipment to the value of £300 which was delivered to them prior to the intervention starting. All schools received the same equipment and were encouraged to make it available to children during non-curricular times

(e.g., recess) and during PE lessons. At the end of the intervention each child received a £10 gift voucher as a gesture of recognition for their engagement in the study.

### Data analysis

Percentage values were calculated for the traffic light criteria to determine which progression zone each one corresponded to. Descriptive statistics were calculated using IBM SPSS (Armonk, NY) for the feasibility and child-level outcomes at T0 and T1. Data provision rates for questionnaire data and accelerometer data were also recorded at T0 and T1. The children's qualitative visual data and recorded transcription data were pooled together for complimentary purposes to add context and depth to the drawings. Credibility was enhanced and researcher biases were reduced by triangulating the two data sources. Verbatim quotes and drawings from the children's participatory focus groups and teachers' semi-structured interviews were extracted to exemplify representation of the participants' experiences and perceptions.

## Results

### Feasibility outcomes traffic light progression criteria

**School and participant recruitment.** Fig 1 presents a CONSORT flow diagram of participants through the study. The six schools that were initially approached agreed to take part in the study, but one withdrew before the study commenced due to staff absences. This meant 83% of the target number of schools were recruited (progression criterion 1 = amber; Table 3). These schools had only one Year 5 class which limited the maximum number of children invited to participate to n = 144. Overall, 54% (n = 108) of the maximum target number of 200 children provided written parental informed consent to take part (progression criterion 2 = amber). This represented a response rate of 75% of the invited children, which ranged from 68% to 93% between the schools. The children were aged 9.6 ± 0.4 years, were predominantly of White British ethnicity (89.8%) with healthy weight status (68.5%), and just over half (51.9%) were girls. In each school one PE teacher and one Year 5 class teacher were recruited (progression criterion 3 = green).

**Intervention delivery, adherence, and acceptability.** The PE teachers delivered 76% of the core MWFG lessons (progression criterion 4 = amber). Across the five schools this reflected 34 lessons taught from a possible 45. Moreover, it accounts for the fact that in one school no lessons were taught following a decision from that school's PE Coordinator which was not communicated to the research team. Over 75% of the 34 lessons were attended by 76% of the recruited children (progression criterion 5 = green). The MWFG programme was perceived as enjoyable by 86% of the children, with no systematic differences in demographic characteristics among the few children who were unsure about or who did not enjoy the programme. Similarly, 83% of the PE teachers and 90% of the class teachers found the programme to be straightforward to deliver and engaging for the children (progression criterion 6 = green).

**Data collection acceptability and feasibility.** Ninety-one percent of the recruited children reported that the data collection sessions were enjoyable. This was corroborated by 80% of the class teachers who observed that data collection worked well and was interesting and engaging for the children (progression criterion 7 = green). At T0, secondary outcome measures were completed by between 80% and 100% of recruited children (progression criterion 8 = green). Data attrition at T1 ranged from 3% (mental health questionnaires) to 35% (accelerometers) (progression criterion 9 = amber). The next highest level of attrition was 12% for the motor competence and anthropometric measures.

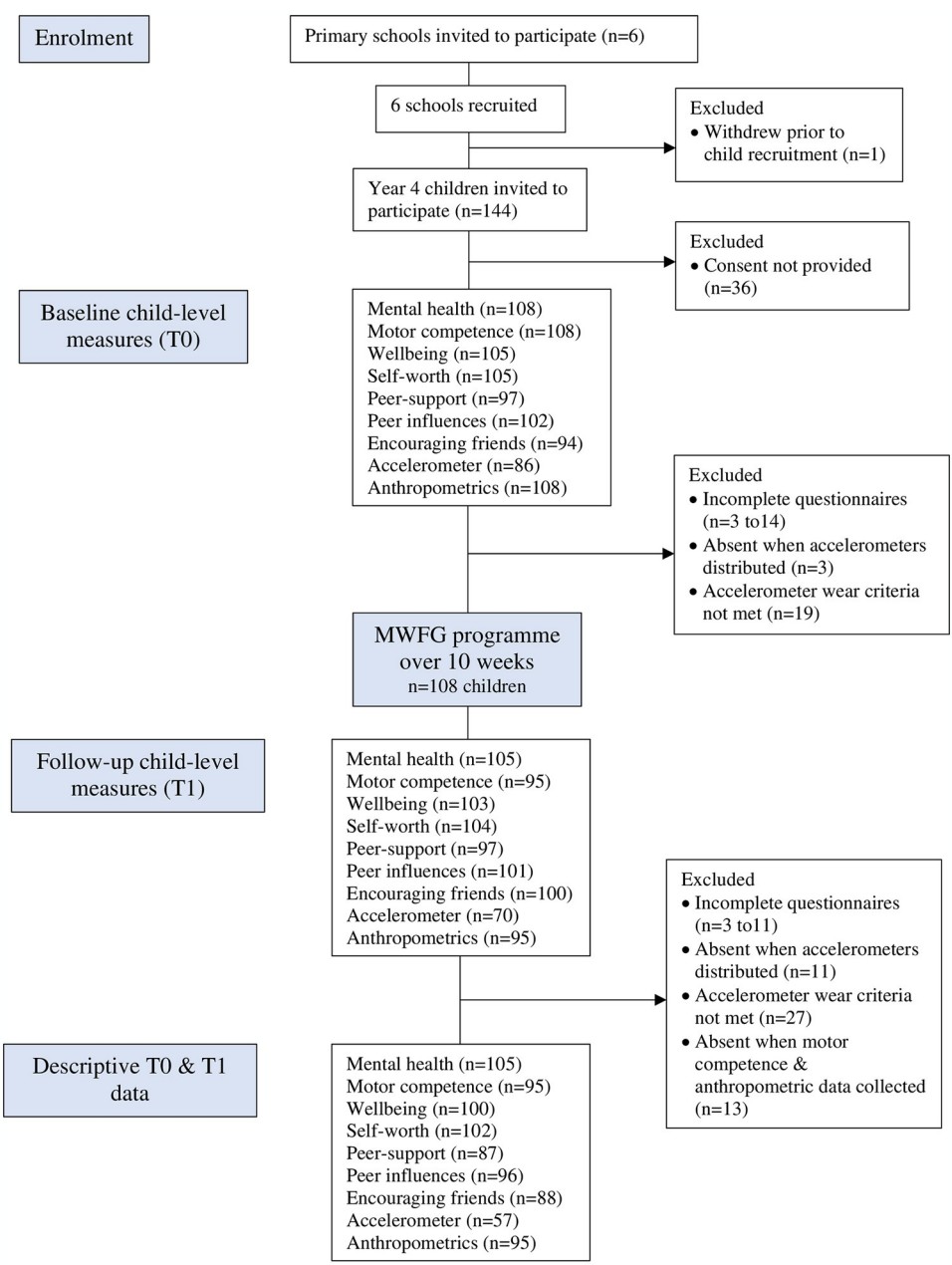

**Fig 1. Flow of participants through the Move Well, Feel Good feasibility intervention (based on CONSORT flow diagram).**

## Qualitative acceptability and feasibility results

Thirty children participated in the write, draw, show, tell participatory focus groups. When asked to write or draw the most memorable activities throughout MWFG, the children most frequently drew the motor competence assessment activities from the data collection sessions. This was consistent across the five focus groups. Further, the points that the children most consistently referred to were fun and enjoyment of the data collection activities themselves and feeling like they were able to both tell the truth and express their feelings when completing

**Table 3. Summary of traffic light feasibility progression criteria results.**

| Progression criteria | Results | Progression decision |
|---|---|---|
| 1. School recruitment (targeting n = 6) | 6 schools recruited (1 withdrew) leaving 5 for the duration of the study. No large (i.e., 2-class entry) schools were recruited, which limited the maximum number of invited children. | 83% of target number of schools recruited. **Amber: Discuss and amend.** |
| 2. Child participant recruitment (targeting n = 200) | 108 of the targeted 144 (75%) children consented and recruited. | 54% of target number of children recruited. **Amber: Discuss and amend.** |
| 3. Deliverer recruitment | 1 class teacher and 1 PE teacher in each of the recruited schools. | 100% of target number of deliverers recruited. **Green: Go.** |
| 4. Intervention dose | Across the 5 schools, 34 lessons out of a possible 45 were delivered over the 9-week programme. No lessons were delivered in 1 school following intervention from the PE Coordinator. | 76% of lessons delivered. **Amber: Discuss and amend.** |
| 5. Intervention adherence | 108 children per week attended a total of 34 from a possible 45 MWFG PE lessons across the 5 schools. This includes the school where no lessons were delivered. | 76% of recruited children attended ≥75% of lessons. **Green: Go.** |
| 6. Acceptability of intervention | 90 children reported that the programme was enjoyable and they learned new skills. 3 PE teachers (2 did not respond) and all 5 class teachers reported that the programme and resources were straightforward to follow and use, and that the programme was interesting and engaging. | 86% of recruited children found the programme acceptable. 83% of PE teachers and 90% of class teachers found the programme acceptable (87% combined). **Green: Go.** |
| 7. Acceptability of data collection methods | 95 children reported that the data collection sessions were enjoyable. All 5 class teachers reported that the data collection methods worked well and children found them interesting and engaging. | 91% of recruited children found the data collection methods acceptable. 80% of class teachers found the data collection methods acceptable. **Green: Go.** |
| 8. Secondary outcome data collected at baseline | Mental health (n = 108) Motor competence (n = 108) Wellbeing (n = 105) Self-worth (n = 105) Peer-support (n = 97) Peer influences (n = 102) Encouraging friends (n = 94) Accelerometer (n = 86) Anthropometrics (n = 108) | 80%-100% of children completed secondary outcome measures at baseline. **Green: Go.** |
| 9. Follow-up secondary outcome data attrition | Mental health; attrition = 3% Motor competence; attrition = 12% Wellbeing; attrition = 5% Self-worth; attrition = 4% Peer-support; attrition = 10% Peer influences; attrition = 6% Encouraging friends; attrition = 7% Accelerometer; attrition = 35% Anthropometrics; attrition = 12% | 65%-97% of children completed secondary outcome measures at baseline. Attrition rate ranged from 3%-35% across the measures. **Amber: Discuss and amend.** |

the questionnaires. An example of the visual (Fig 2) and spoken data are presented to illustrate this.

> "So, I actually drew the whole thing [motor competence assessment/practical skills circuit]. . .I think that's probably the most fun thing we did [CAMSA and Dragon Challenge] because all the things you had to do are one. I did not like the scale [anthropometric measures] because it was very, very boring"

[Focus group participant, School 2].

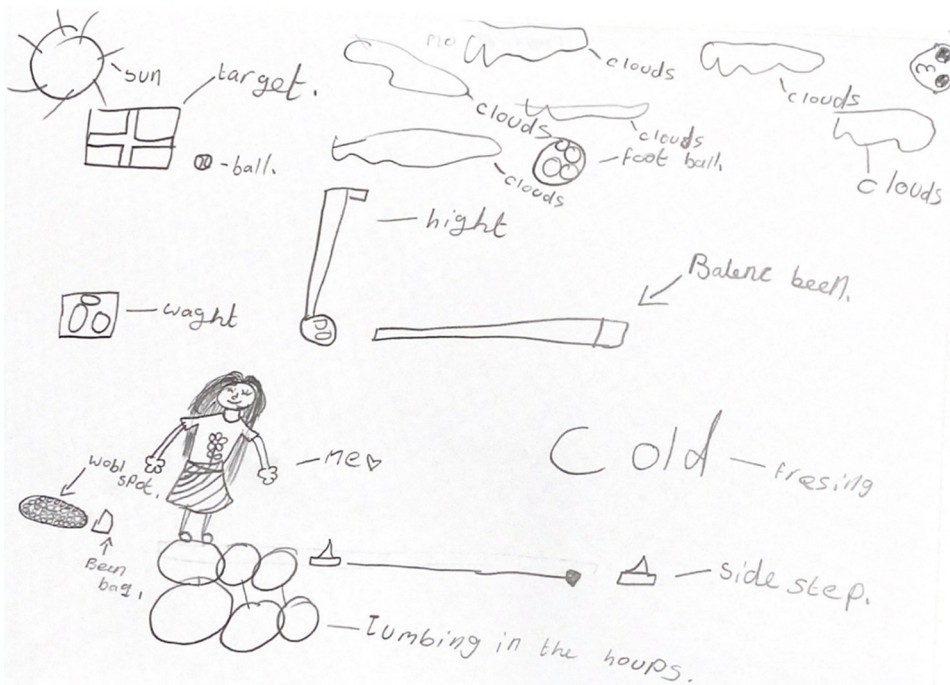

**Fig 2. Drawing from a boy aged 10, illustrating the activities that were memorable to him throughout the MWFG programme.**

"I liked the survey because when it's a piece of paper asking you the questions you're more likely to tell the truth. . .so it's nice to let other people know without having to say anything."

[Girl, School 3]).

All children spoke positively about the motor competence assessments (e.g., "I really liked the ones with the jumping in the hoops one-footed, and the balancing spot" (Boy—School 1), with mixed comments about the anthropometric measures. There were also some varied comments about the accelerometers ("The watch kind of irritated me, but it was good" (Girl–School 1)), although most children spoke positively about these and enjoyed the novelty of wearing them.

The teachers' interview responses typically mentioned the MWFG teaching resources, engagement of the children, the timing of the intervention, delivery constraints and data collection methods. Teachers commented positively on the volume, detail, and clarity of the teaching resources and that the content was appropriate for the children's abilities (e.g., ". . .some of them have got it straight away and some it took a little bit longer, but I don't think there was anyone that really, really struggled with the activities that were there." [PE specialist, male, School 4]). Importantly, the MWFG lesson activities were deemed to be inclusive and engaging for the children. One teacher stated, "I've got some children who are very reluctant to take part in PE. . .. I thought it [the programme overall?] was great because

a lot of those children who do struggle really enjoyed the programme and really wanted to take part, and it was fun."

[Class teacher, female, School 4].

Teachers felt that the 10 weeks duration of MWFG was appropriate, but that the timing in the first term back after the summer break was sometimes problematic. This was partly because of how busy that period was (e.g., "I would never, ever do it in the autumn term ever again because it is literally the worst term to do it in." [Class teacher, female, School 4]) and partly because of child absences and competing school priorities in December (e.g., festive celebrations). Moreover, timing issues were sometimes compounded by limited availability of indoor practical spaces for MWFG lesson delivery or data collection (e.g., ". . .every time you've emailed saying 'when can we come?', the reality's been, there isn't a space to do that. So yes, that has been quite tricky, really."

[Class teacher, female, School 5]).

The teachers were generally positive about the data collection methods. In particular, references were made to the accelerometers (e.g., ". . . [children] absolutely loved the watches. . ." [Class teacher, female, School 4]) and questionnaires (e.g., ". . .there's definitely calming effects with them critically thinking about themselves, noticing themselves, noticing how they're feeling. . .the questionnaire, to evaluate yourself, can have that effect. . ." [Class teacher, female, School 2]). In contrast, one teacher mentioned how some children were reluctant to be weighed and whether additional privacy measures could have been taken in these situations ("I have got a few [pupils] who are very self-conscious, because obviously they are overweight for their age at this point. . .but maybe it would have been better to take them. . .just a bit further away from the others. . ."

[Class teacher, female, School 4]).

## Child-level outcomes

Fifty-six girls and 52 boys (mean age = 9.6 ± 0.4 years) took part in the study. Mean BMI z-score was 0.69 ± 1.4; 68.5% of the children were categorised as healthy weight, and the remaining 31.5% were in the overweight/obese category. On average, the children resided in areas of relatively high deprivation (mean EIMD decile = 3.6 ± 3.0). Descriptive results for the secondary outcomes at T0 and T1 are detailed in Table 4. At T1 Compared to T0, motor competence scores for the CAMSA and Dragon Challenge stability skills were 13.7% and 38.3% higher, respectively. The narrowness of the 95% confidence intervals for the T1-T0 differences suggested that these changes were meaningful. Favourable changes in mental health outcomes were also observed between T0 and T1, with meaningful differences indicated for SDQ-assessed total difficulties, externalising difficulties, and prosocial behaviour. Wellbeing and psychosocial outcome scores were generally higher at T1 but the differences seen were negligible. Children wore the accelerometers for 385 min·school-day$^{-1}$ (99.7% of total school day duration) on 3.9 (T0) and 4.0 (T1) school-days per week. Children were less active and more sedentary at T1 than at T0 when the expected seasonal changes in weather conditions were also observed. Specifically, between T0 and T1 the average ambient day-time temperature decreased from 13.3˚C to 3.8˚C, and rainfall increased from 1.7 mm·day$^{-1}$ to 3.1 mm·day$^{-1}$.

**Table 4. Descriptive comparison of mean (SD) secondary outcomes at T0 and T1.**

| Outcome | *n* | T0 | T1 | T1-T0 mean difference (95% CI) |
|---|---|---|---|---|
| *Motor competence* | | | | |
| CAMSA overall score | 95 | 29.91 (8.05) | 34.01 (7.27) | **4.11 (2.95, 5.26)** |
| DC stability skills score | 95 | 4.59 (2.51) | 6.35 (2.42) | **1.76 (1.25, 2.26)** |
| *Mental health* | | | | |
| MMF: emotional difficulties | 105 | 6.46 (4.10) | 6.31 (4.13) | -0.15 (-0.54, 0.82) |
| MMF: behavioural difficulties | 105 | 2.42 (2.40) | 2.10 (2.27) | -0.32 (-0.13, 0.77) |
| SDQ: total difficulties | 105 | 12.42 (6.98) | 11.33 (6.90) | **-1.09 (-0.10, -2.07)** |
| SDQ: externalising difficulties | 105 | 5.96 (3.91) | 5.30 (3.74) | **-0.66 (-1.21, -0.11)** |
| SDQ: internalising difficulties | 105 | 6.43 (4.19) | 6.02 (4.14) | -0.40 (-0.21, 1.06) |
| SDQ: prosocial behaviour | 105 | 7.92 (2.17) | 8.52 (1.69) | **0.60 (0.26, 0.94)** |
| *Wellbeing* | | | | |
| KIDSCREEN-10: T-scores | 100 | 50.12 (9.61) | 50.48 (11.10) | 0.37 (-2.23, 1.50) |
| *Psychosocial outcomes* | | | | |
| Global self-worth | 102 | 3.28 (0.70) | 3.25 (0.75) | 0.03 (-0.11, 0.16) |
| Support from other children | 87 | 3.92 (0.91) | 3.80 (0.89) | -0.12 (-0.03, 0.28) |
| Social influences on physical activity | 96 | 13.43 (2.28) | 13.74 (2.28) | -0.31 (-0.85, 0.22) |
| Encourage friends' physical activity | 88 | 1.22 (0.65) | 1.24 (0.63) | 0.02 (-0.18, 0.14) |
| *School-day physical activity outcomes* | | | | |
| School-day duration (min·day$^{-1}$) | | 385.9 (6.6) | 385.9 (6.6) | |
| Number of valid days | 57 | 3.91 (0.29) | 3.96 (0.53) | -0.05 (-0.22, 0.12) |
| Wear time (min·day$^{-1}$) | 57 | 385.20 (7.20) | 384.60 (8.40) | 0.48 (-0.60, 1.50) |
| SED (min·day$^{-1}$) | 57 | 227.01 (27.87) | 243.27 (30.03) | **16.26 (26.58, 5.95)** |
| LPA (min·day$^{-1}$) | 57 | 158.06 (31.95) | 138.84 (32.22) | **-19.22 (-7.58, -30.86)** |
| MPA (min·day$^{-1}$) | 57 | 29.36 (8.95) | 25.43 (9.49) | **-3.93 (-6.86–1.00)** |
| VPA (min·day$^{-1}$) | 57 | 4.88 (2.36) | 4.40 (3.10) | 0.48 (-0.30, 126) |
| MVPA (min·day$^{-1}$) | 57 | 34.24 (10.75) | 29.83 (12.25) | **-4.41 (-8.02, -0.81)** |
| Average acceleration (mg) | 57 | 80.04 (16.07) | 70.81 (20.12) | **9.23 (3.30, 15.20)** |
| Intensity gradient | 57 | -1.81 (0.15) | -1.89 (0.16) | **0.08 (0.04, 0.12)** |

Note. Bold font highlights 95% confidence intervals not crossing zero which suggests meaningful T1-T0 differences; CAMSA: Canadian Agility and Movement Skill Assessment; DC: Dragon Challenge; MMF: Me and My Feelings questionnaire; SDQ: Strengths and Difficulties Questionnaire; SED: sedentary time; LPA: light physical activity; MPA: moderate physical activity; VPA: vigorous physical activity; MVPA: moderate-to-vigorous physical activity.

## Discussion

### Feasibility outcomes

This study evaluated the feasibility and acceptability of the MWFG intervention and also reported changes in child-level outcomes. Regarding the nine traffic light feasibility criteria for progressing to a pilot trial, five were fully met (i.e., green = go). These related to deliverer recruitment (100% of target number), intervention adherence (76% of children attended lessons), intervention acceptability (86%, 83%, and 90% of children, PE teachers, and class teachers, respectively), acceptability of data collection methods (91% of children and 80% of class teachers, respectively), and secondary outcome data collected at baseline (measures completed by 80%-100% of children). The remaining progression criteria were partially met (i.e., amber = discuss and amend) and were related to school recruitment (83% of target number),

child recruitment (54% of maximum target number), intervention dose (76% of lessons delivered), and follow-up secondary outcome data attrition (T1 measures completed by 65%-97% of children). Of the child-level outcomes, favourable differences in motor competence, mental health, and prosocial behaviour were observed.

School recruitment was initially 100% until one school withdrew shortly after initially agreeing, due to multiple staff absences, a common reason for attrition in school-based research studies. Jago and colleagues suggested the need for increased engagement at the school recruitment stage to better understand school staffing capacities to help avoid such situations [60]. We recruited 54% of the maximum target number of 200 children. This maximum was assumed from that of our previous work in the same locality [61], namely that we would recruit at least two 'large sized' schools (i.e., those with at least two Year 5 classes). This assumption was not realised as all schools that agreed to take part had one Year 5 class, which limited the maximum target sample of children to n = 144. A replacement school was not sought because at the time it was felt that extending the duration of school recruitment would place pressure on the timeline for the co-production phase. As a result, this child recruitment criterion did not meet the 'green/go' criterion of ≥75% children. To an extent, this lower recruitment rate was offset by the high proportion of targeted children providing parental/carer consent to take part (75%), which was greater than that reported in other UK primary school motor competence/physical activity feasibility interventions [60, 62]. Notwithstanding this favourable participation rate, the 'opt-in' method of parental/carer informed consent that we used likely inhibited the potential to recruit more children. Our previous work with schools in the same location which was ethically approved to use 'opt-out' consent returned a 95% participation rate [61]. An opt-out consent approach reduces burden on researchers, teachers, and parents, yields higher response rates [63], and reduces selection bias [64]. Moreover, when an opt-out approach is adopted, parental/carer consent is supplemented by school management, which acts in *loco parentis*, and provides a further ethical safety net [65]. To progress MWFG to a pilot trial that achieves target recruitment levels within planned timescales, a convincing case for opt-out consent to the institutional ethics committee would be made.

Thirty-four MWFG PE lessons were taught from a possible 45 (76% of scheduled lessons), which did not meet the ≥80% green/go progression criterion. In one school two lessons were not delivered because the PE teacher was absent due to injury. Of greater significance was the decision by another school not to deliver any of the MWFG lessons. Instead, the PE teacher was instructed by the more senior PE Coordinator to teach the school's regular PE curriculum, and this decision was only communicated to the researchers by the PE teacher during the end of programme interviews. This PE Coordinator had taken part in the phase 1 co-creation workshops and this decision was not anticipated, particularly without any consultation with the research team whilst the study was ongoing. Notwithstanding this specific situation, it is important to recognise that schools are fluid and dynamic environments where unpredicted issues can arise, such as here [66]. Further, teachers have a relatively high degree of professional autonomy as well as diverse responsibilities (e.g., subject coordination, leadership roles) [67]. As a result, teachers also have many competing priorities, and in most cases will view involvement in research projects as that of additional workload [66]. We can thus only speculate that the decision not to allow the MWFG lessons to be taught was based on a combination of factors, and that time pressures to meet other demands may have taken priority over the need to discuss matters with the research team.

The factors causing these three amber progression criteria results could have been temporary and were, arguably, remediable [37]. For example, the school withdrawing due to staff illness and child recruitment being below the maximum target number could have been remedied by extending the school recruitment phase and the geographical location to ensure

that schools with more than one Year 5 class were recruited. As such, this may have delayed the start of the co-creation phase, but on reflection there was some flexibility to adapt the timings of the co-creation workshops. The third amber criteria that related to one school not delivering the MWFG lessons could have been remedied by a discussion between the school and research team to agree a resolution. Avery et al. recommend that in such cases where progression criteria fall into the amber 'discuss and amend' zone as a result of remediable and temporary factors, this should not compromise progression to a subsequent trial [37].

The fourth amber criterion related to the 35% accelerometer data attrition at T1. As has been observed in similar studies, loss of data at follow-up was partly due to child absences at the time of T1 data collection (n = 11) [62]. A significant cause however, which has also been reported previously, was that 27 children did not achieve the accelerometer wear criteria [62]. As was the case in MWFG, Johnstone and colleagues' Active Play feasibility intervention collected accelerometer data in September and December. They had significant accelerometer data attrition and suggested that because December is, typically, a congested time in UK primary schools (e.g., end of term assessments, festive celebrations and performances, wet weather for outdoor PE, etc) it may not be deemed suitable to collect accelerometer data in this month [62]. In addition to these factors there was a spate of illness among the children around the time of T1 data collection. One teacher summed this up by stating, "It's Christmas week and you haven't managed to get all the results. . .because of parties and bugs [illnesses] and all of that" [Class teacher, female, School 5]. The accelerometer outcomes were included to gauge whether the MWFG programme influenced school-day movement behaviours, as is inferred in the EESH model. If accelerometers were retained in a pilot trial to provide secondary outcome data, we would need to consider modifications to the wear protocol (e.g., more frequent parental text message reminders to wear the device, less stringent wear time inclusion criteria based on sensitivity analysis, more frequent or higher value incentives). Aside from the accelerometer data the rate of attrition ranged from 3% to 12%, which met the green/go progression criterion of $\leq$25% and is consistent with other school-based physical activity feasibility studies [68, 69].

The teachers' comments relating to the timing of the intervention and data collection in December were supported by comments of others who also highlighted the general busyness of schools during the September to December term. Moreover, the time commitment needed to accommodate the programme was raised in the MWFG teacher interviews (e.g., "It's [data collection activities] just been a massive time commitment, the second we give you [the research team] two afternoons, we've then got to find those two afternoons to catch up our curriculum elsewhere." [Class teacher, female, School 5]. Time available in schools for interventions and for teachers to engage with them is indeed arguably the most significant barrier to successful project implementation in this setting [66].

The other five progression criteria achieved green/go status, which reflected high levels of intervention acceptability in respect of children's attendance and enjoyment of lessons and engaged participation in the research data collection. One child noted that the CAMSA and Dragon Challenge motor competence assessments were ". . .probably the most fun thing we [pupils] did because it had so many different fun activities in it" [Boy, School 2]. Some children also valued completing the mental health questionnaires as a way of anonymously expressing their feelings.

Of note were the findings that PE and class teachers were recruited in all schools, and 87% of all teachers agreed that the MWFG lesson resources were easy to use and interesting for the children. One PE specialist stated, "I think the resources were good, it had all the details in it that you needed. I think the kids received it quite well, they enjoyed doing different activities and the games" [PE specialist, male, School 4]. This view was consistent with those of the class

teachers (e.g., "It [MWFG lesson resource pack] was good being cross-curricular, I was pleasantly surprised how much it referenced and touched on other subjects, like working together as a team for PSHE [Personal, Social, and Health Education]. I think anyone could pick up that folder, read it and use it as it's intended to be used" [Class teacher, male, School 2]). The PE teachers particularly appreciated how the lesson plans contained embedded electronic links to the activity cards, which facilitated more efficient lesson delivery indoors and outside. One PE teacher commented that,

> "The resources were perfect. You could put the cards up on the screen so the pupils could see, so there's a demonstration, a picture, and a step-by-step rather than just listening to me and me showing it, it was good to have the points where pupils could read to each other as well"

> [PE specialist, male, School 1 and 3],

and that,

> "It was helpful when you put the links together for me, so I didn't have to go backwards and forwards for the file. It was a lot easier just to click the link and I'd know where I was working then using the iPad outside"

> [PE specialist, male, School 1 and 3].

However, improvements to some of the resources were also suggested, such as the inclusion of laminated activity task cards for outdoor use to complement the electronic and paper versions.

Child engagement and quality of programme resources are important indicators of intervention feasibility as high lesson attendance is vital to children's intervention participation and learning [60], and lesson resources need to be simple and straightforward to use [68], particularly when additional teacher lesson preparation time may be not be possible [70]. Although MWFG was primarily delivered through PE lessons the lesson content was deliberately task-focused emphasising motor skill development through guided practice and progressive activities, which often involved peer learning. In some schools this was a welcome departure from a more traditional sport-focused PE curriculum and may have benefited a wider range of children. This point was summarised by a teacher who said,

> "I feel like, for me, kids who do struggle with the sport side of things, it's been great, because they've actually been able to have some success, whereas normally, if we're doing anything like footie [football/soccer], they can't cope because they know they can't do it"

> [Class teacher, female, school 4].

The intention of the write, draw, show, tell task was to act as a stimulus to get the children to engage in the focus groups. We did not intend to take a consultative approach with the children regarding the data collection process, but it was enlightening to see the data that emerged which we feel was valuable from a feasibility and acceptability standpoint. Combining the visual and verbatim data enhanced data credibility, and revealed complementary findings on children's views, experiences and perceptions of the data collection process which were not captured in the quantitative feasibility survey. For example, an activity might have been memorable to a child because of its negative experience (e.g., anthropometric measures considered 'boring'), but without the supporting verbatim data to provide the context this would not have

been acknowledged. This further emphasises the utility of using multiple qualitative techniques with children rather than relying on one method. alone [71]. Pooling multiple data sources together reduced the risk of misinterpretation by researchers, provided greater depth and context to the visual data and enhanced confidence in the findings [71]. Further, the anonymised qualitative findings regarding the data collection methods could be relayed back to schools during the recruitment phase in future school-based research, to demonstrate the feasibility and acceptability of such methods from the perspectives of both the children and teachers in their own words.

## Child-level outcomes

Descriptive analysis of motor competence and mental health outcomes as the respective key EESH model exposure and outcome variables revealed favourable changes in the hypothesised directions. The T1-T0 change scores for overall CAMSA (assessing locomotor and object control skills) (+13.7%) and Dragon Challenge stability skills (+8.1%) and the narrowness of the mean change confidence intervals suggested that the changes were meaningful. This was reflected in the proportion of children advancing between the CAMSA motor competence categories of 'Beginning', Progressing', Achieving', and 'Excelling' at T0 and T1 (S6 File). Further, for a 10-year old child the mean CAMSA change score of 4.11 would reflect progression from the 'Beginning' motor competence category to the 'Progressing' category, while for a child with a score around the middle range of the 'Progressing category', a 4.11 improvement would progress them into the 'Achieving' or 'Excelling' categories [72]. Similarly, in relation to the SDQ total difficulties score, for a child at the lower end of the UK normative categories of 'Very high', 'High', and 'Slightly raised' risk for overall mental health difficulties, the observed -1.09 change (-8.8%) would be sufficient to move them down to the next risk category [43]. These favourable changes in motor competence and mental health are consistent with the ESSH, but to determine whether the findings are supportive of the model would require a structural equation model applied to a larger dataset.

We used the SDQ and shorter MMF tools to assess mental health but the magnitudes of the relative T1-T0 changes were much smaller for MMF. It is unclear why the changes differed between the tools to the extent that they did. The more established SDQ includes more items than MMF and was used to assess MMF's construct validity [44]. Nevertheless, MMF has previously demonstrated that it can capture clinical mental health need to a similar level as the SDQ [44], although in a community sample, detection of scores above a clinical threshold was lower than that reported for the SDQ [73]. Our motivation for including both mental health measures was to assess their acceptability among children and teachers, as well as to describe changes in children's responses between T0 and T1. One child felt that some questions on the MMF "...don't make sense" [Boy, School 2], but this was countered by other children who were positive about the questionnaires (e.g., "...I understood all the questions and I liked it" [Boy, School 3]). Overall, consistent negative feedback about either tool was not received, inferring that both were broadly considered as acceptable. Moreover, the Cronbach's alpha inter-item reliability scores for both mental health questionnaires exceeded 0.80, suggesting that on balance the children understood the questions and responded to them in a consistent manner. MMF has fewer items than the SDQ and scores are distilled down to separate emotional and behaviour difficulties constructs, rather than to a score for overall mental health, which could possibly be more sensitive to change. Further, the low mean SDQ total difficulties scores indicated that the children were generally at low risk of poor mental health [43]. A combination of these factors and the relatively high children's mental health scores may have limited the potential magnitude of the MMF's T1-T0 change scores.

With the exception of the prosocial behaviour construct of the SDQ, we observed negligible changes in psychosocial outcomes. The individual SDQ prosocial items focused on children being considerate of others' feelings, sharing with others, being helpful, and kind. These qualities reflected the underlying psychosocial qualities of MWFG that were embedded into the PE lesson plans and classroom activity ideas. It is possible that these SDQ prosocial behaviour items represented a more suitable measure of the intended psychosocial development in MWFG, than the stand-alone questionnaires that focused on self-worth, social influences and support from others, and encouragement of friends' physical activity. If this were the case, solely using the SDQ to assess mental health and prosocial behaviour would be significantly less burdensome for children, schools, and researchers than employing multiple measures as we did in this feasibility study.

Differences in school-day sedentary time and physical activity outcomes were observed between T0 and T1. These outcomes were included in the study because physical inactivity is positioned in the EESH model as a factor influenced by poor motor competence that has a bi-directional relationship with obesity risk, and which influences the psychosocial mediators of mental health [18]. It is possible that physical inactivity and sedentary time were positively affected during discrete periods of the school-day, such as the MWFG PE lessons or classroom lessons. However, we did not analyse accelerometer data for specific lesson timings, and even if favourable changes did occur during these lessons, it is also possible that compensation effects could occur elsewhere during the school-day. The most likely explanation for the observed changes in movement behaviours was the seasonal differences in weather conditions between September and December, which were reflected by a 71% drop in temperature and 82% increase in rainfall between T0 and T1. Indeed, a recent meta-analysis has demonstrated how children's physical activity and sedentary behaviours are adversely affected by unfavourable weather conditions [74]. In a MWFG pilot trial, analysis of physical activity data would be adjusted to account for weather conditions.

An important strength of this study was the nine feasibility progression criteria which guided our assessment of feasibility and acceptability. We developed the intervention content through co-creation with children, teachers, and PE teachers [31], which positively reflected participants' levels of engagement. Use of mixed methods that employed validated quantitative measures and developmentally-appropriate qualitative approaches afforded children and teachers opportunities to express their views and perceptions in different ways. Moreover, the intervention was conceptually underpinned by the ESSH model, which informed the choice of child-level outcome measures. Overall, the design and implementation of the study (e.g., deliverer selection, prescribed intervention delivery, etc) limited the 'risk of generalisability biases' [75]. Limitations of the study were the lack of comparison group, relatively short duration, and mixed-quality of teacher-researcher communication which resulted in non-delivery of the intervention in one school. Further, a small number of schools and participants participated which, although typical of feasibility studies, also limits the generalisability of the findings.

## Conclusions

Five of the nine progression criteria were green, while the remaining amber criteria that related to school and child recruitment and intervention delivery were all remediable. These areas could be addressed during the recruitment period by targeting a larger pool of schools, using opt-out consent, improving researchers' understanding of staffing capacities and overloaded periods of the school year, and by reinforcing the importance of early communication between schools and researchers. Moreover, favourable changes were observed in children's motor competence, mental health, and prosocial behaviour. MWFG is therefore an acceptable and

feasible school motor competence intervention to promote positive mental health. With specific content and delivery modifications it could be progressed to a pilot trial using a more robust design.

## Supporting information

**S1 File. Example lesson plan.**
(PDF)

**S2 File. Example non-curricular skill snack activity card.**
(PDF)

**S3 File. Detailed description of feasibility outcome measure methods and administration.**
(PDF)

**S4 File. TREND checklist.**
(PDF)

**S5 File. Ethics-approved protocol.**
(PDF)

**S6 File. Proportion of children in each CAMSA category.**
(PDF)

## Acknowledgments

The authors gratefully acknowledge the engagement and support of all the participating children and teachers, and staff at West Lancashire Sport Partnership.

## Author Contributions

**Conceptualization:** Stuart J. Fairclough, Lawrence Foweather, Lynne M. Boddy, Richard Tyler.

**Data curation:** Stuart J. Fairclough, Lauren Clifford.

**Formal analysis:** Stuart J. Fairclough, Lauren Clifford, Zoe R. Knowles.

**Funding acquisition:** Stuart J. Fairclough, Lawrence Foweather, Zoe R. Knowles, Lynne M. Boddy, Emma Ashworth, Richard Tyler.

**Investigation:** Stuart J. Fairclough, Lauren Clifford, Richard Tyler.

**Methodology:** Stuart J. Fairclough, Lawrence Foweather, Zoe R. Knowles, Richard Tyler.

**Project administration:** Stuart J. Fairclough, Lauren Clifford.

**Supervision:** Stuart J. Fairclough.

**Validation:** Lauren Clifford, Zoe R. Knowles.

**Visualization:** Stuart J. Fairclough.

**Writing – original draft:** Stuart J. Fairclough, Lauren Clifford, Richard Tyler.

**Writing – review & editing:** Stuart J. Fairclough, Lauren Clifford, Lawrence Foweather, Zoe R. Knowles, Lynne M. Boddy, Emma Ashworth, Richard Tyler.

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
