## [Decision Letter · Decision Letter 0]

21 Feb 2024

PONE-D-23-31624Move Well, Feel Good: Feasibility and acceptability of a school-based motor competence intervention to promote positive mental healthPLOS ONE

Dear Dr. Fairclough,

Thank you for submitting your manuscript to PLOS ONE. After careful consideration, we feel that it has merit but does not fully meet PLOS ONE’s publication criteria as it currently stands. Therefore, we invite you to submit a revised version of the manuscript that addresses the points raised during the review process.

We look forward to receiving your revised manuscript.

Kind regards,

Henri Tilga, PhD

Academic Editor

PLOS ONE

Journal Requirements:

2. Thank you for stating the following financial disclosure: "This work was supported by a grant from The Waterloo Foundation (#1669/4195) that was awarded to SJF, RT, LF, LMB, ZK, and EA."

Additional Editor Comments :

The Reviewers have provided several useful comments to increase the quality of this manuscript. Please carefully follow all the comments made by the Reviewers and revise the manuscript accordingly.

Reviewers' comments:

Reviewer's Responses to Questions

**Comments to the Author**

1. Is the manuscript technically sound, and do the data support the conclusions?

Reviewer #1: Yes

Reviewer #2: Yes

Reviewer #3: Yes

2. Has the statistical analysis been performed appropriately and rigorously? 

Reviewer #1: Yes

Reviewer #2: N/A

Reviewer #3: Yes

3. Have the authors made all data underlying the findings in their manuscript fully available?

Reviewer #1: No

Reviewer #2: Yes

Reviewer #3: Yes

4. Is the manuscript presented in an intelligible fashion and written in standard English?

Reviewer #1: Yes

Reviewer #2: Yes

Reviewer #3: Yes

5. Review Comments to the Author

Reviewer #1: The present paper describes a single-arm feasibility trial. A point must be noted feasibility is the study is not a powered trial. It should be used to evaluate the feasibility of recruitment, randomization, retention, assessment procedures, outcome selection, new methods, and implementation of the novel intervention to name a few. As a result, I recommend removing the power analysis that is reported in line 116-124. Since testing of hypothesis is never the aim of this type of trial, no power analysis, and sample size using such analysis is needed. Look at the guidance of Leon et al. (Journal of Psychiatric Research 45 (2011) 626e629).

Another point I want to note, the ultimate goal of such a feasibility/pilot study is to collect information in preparation for a large-scale powered trial. As such a brief section outlining “lessons learned” and changes to be made in future studies will be helpful for the curious reader.

Reviewer #2: Thank you for the opportunity to review the manuscript entitled” Move Well, Feel Good: Feasibility and acceptability of a school-based motor competence intervention to promote positive mental health”. This manuscript addressed an important topic that is children intervention to foster their motor competence and psychosocial skills in order to promote their health. It was my pleasure to read this paper because it is really related with my research field and I am familiar with most of the instruments used. This article is well-written and clear, and most of my doubts were solved as I was reading the document. However, there are some aspects that should be improved prior to consider for publication. Below you can see my suggestions. I hope these comments could help the authors to improve the quality of their manuscript.

Comment 1. In my humble opinion there is a lack of flow in the introduction. The first time I read the paper I did not understand the mention to the lockdown at the beginning of this section as no information about the lockdown was included in the abstract. Even when I noticed the relationship with the data collection and study variables, I am wondering if the authors could include information about this link in the abstract. Additionally, I think that the authors could reorganize the introduction and include first the information about the Elaborated Environmental Stress Hypothesis model (EESH) (lines 72-81) and after that, the information regarding the lockdown (lines 52-70). Perhaps this strategy could help them to link it better with the second last paragraph of the introduction. This is only a suggestion.

Comment 2. As some time has passed since lockdown, I think it would be interesting to include references that support the idea that the effects derived from lockdown still persist.

Comment 3. The recruitment was deeply explained however I appreciate if the authors could add concrete information about number of participants, schools, mean age, gender, etc.

Comment 4. There was a control of the voluntary/optional activities? (The authors stated “which were supplemented by optional class-time, break-time, and home activities”). If yes, please explain it in detail.

Comment 5. Only three stability skills from the dragon challenge (DC) were used? How do you compute a motor quotient for these three tasks? Please, explain it in detail. Why the authors did not consider other alternatives such as KTK? In any case, I consider the authors could examine the correlations between CAMSA and DC (3 tasks) to see if both measures are related (as both are supposed to measure participants’ motor competence).

Comment 6. Could you provide further information about the validity and reliability of both tests? I strongly recommend to explain if there was a training period of raters and comparison between them and experts.

Comment 7: I really would like to know, as a reader, which questions the authors/research team asked to obtain data presented in lines 387-395.

Comment 8: Could you further explain how the focus groups were conducted and recorded? (in which room, if silence was required, etc.).

Comment 9: I really appreciate if the authors could include examples of this result: “All children spoke positively about the motor competence assessments” (line 423).

Comment 10: Could you provide further statistical results such as mean comparisons (e.g., paired t-test) between T0 and T1? This information could complement the results included in table 4.

Comment 11: Have the authors check the gender effect? As psychosocial variables are included, I am wondering if changes can be done only in boys or girls. I suggest the authors to consider a model gender*time to see the effects of the MWFG.

Comment 12: In lines 656-658, the authors stated: “The T1-T0 change scores for overall CAMSA (assessing locomotor and object control skills) (+13.7%) and Dragon Challenge stability skills (+8.1%) and the narrowness of the mean change confidence intervals suggested that the changes were meaningful”. I think that authors should consider my proposal of comments 10 and 11 as they could made assertions based on statistical results.

Comment 13: In lines 659 and 660, the authors mentioned the categories of the CAMSA. Could you provide a supplementary material with descriptive information about how many participants are in each category?

Reviewer #3: I am grateful for the opportunity to review this interesting and well written manuscript. The primary purpose of the study is to evaluate the feasibility and acceptability of Move Well, Feel Good (MWFG). Feasibility and acceptability are fundamental to the field of implementation research, and are here examined exemplary well, which leads to results that provide good information about the qualities of the intervention in these areas. If possible, I would like to see some further problematization of the results, for example regarding what characterizes the 14% of the children who did not perceive the program as fun. That kind of information would further enhance this already well-conducted and well-reported study, I therefor recommend a minor revision including these aspects in the manuscript. .

6. PLOS authors have the option to publish the peer review history of their article (what does this mean?). If published, this will include your full peer review and any attached files.

Reviewer #1: No

Reviewer #2: No

Reviewer #3: No

---

## [Author Response · Author response to Decision Letter 0]

20 Mar 2024

Please see uploaded 'Responses to Reviewers' document for these responses in table format.

R1

Comment number Comment Response

Reviewer 1 

1 The present paper describes a single-arm feasibility trial. A point must be noted feasibility is the study is not a powered trial. It should be used to evaluate the feasibility of recruitment, randomization, retention, assessment procedures, outcome selection, new methods, and implementation of the novel intervention to name a few. As a result, I recommend removing the power analysis that is reported in line 116-124. Since testing of hypothesis is never the aim of this type of trial, no power analysis, and sample size using such analysis is needed. Look at the guidance of Leon et al. (Journal of Psychiatric Research 45 (2011) 626e629). Thank you for this comment. We agree that our feasibility trial was not powered to evaluate effectiveness of the child-level secondary outcomes, and this was never the intention. This was reflected in the choice of sample size estimation that was used, which related only to the feasibility outcomes, which were the primary outcomes in our study. The method of Lewis et al. [1] is based on a traffic light system of feasibility outcomes which are expressed as progression criteria. This method does not seek to estimate a sample size for the purposes of testing a hypothesis of intervention effectiveness. To make this clearer we have extended the sub-heading and added more detail to the opening sentence (lines 147-151). This now reads: “Sample size estimation for feasibility outcomes. Using pilot study progression criteria Red/Stop upper limit and Green/Go lower limit reference tables specifically develop for feasibility outcomes (28) we estimated a sample size for child recruitment.”

2 Another point I want to note, the ultimate goal of such a feasibility/pilot study is to collect information in preparation for a large-scale powered trial. As such a brief section outlining “lessons learned” and changes to be made in future studies will be helpful for the curious reader. The manuscript is already quite long and although we did initially discuss adding a section on ‘lessons learned’ we ultimately decided against this. This was in the interests of brevity, and also because we felt that the conclusion already highlighted these points in the form of the remediable progression criteria and how they could be addressed (“These areas could be addressed during the recruitment period by targeting a larger pool of schools, using opt-out consent, improving researchers’ understanding of staffing capacities and overloaded periods of the school year, and by reinforcing the importance of early communication between schools and researchers… With specific content and delivery modifications it could be progressed to a pilot trial with a more robust design) (lines 778-786). For these reasons we respectfully thank the reviewer for this suggestion but feel that it is already addressed in the manuscript. 

Reviewer 2 

1 Thank you for the opportunity to review the manuscript entitled” Move Well, Feel Good: Feasibility and acceptability of a school-based motor competence intervention to promote positive mental health”. This manuscript addressed an important topic that is children intervention to foster their motor competence and psychosocial skills in order to promote their health. It was my pleasure to read this paper because it is really related with my research field and I am familiar with most of the instruments used. This article is well-written and clear, and most of my doubts were solved as I was reading the document. However, there are some aspects that should be improved prior to consider for publication. Below you can see my suggestions. I hope these comments could help the authors to improve the quality of their manuscript. Thank you for your positive evaluation of the manuscript.

2 In my humble opinion there is a lack of flow in the introduction. The first time I read the paper I did not understand the mention to the lockdown at the beginning of this section as no information about the lockdown was included in the abstract. Even when I noticed the relationship with the data collection and study variables, I am wondering if the authors could include information about this link in the abstract. 

Additionally, I think that the authors could reorganize the introduction and include first the information about the Elaborated Environmental Stress Hypothesis model (EESH) (lines 72-81) and after that, the information regarding the lockdown (lines 52-70). Perhaps this strategy could help them to link it better with the second last paragraph of the introduction. This is only a suggestion. We have amended the abstract which now includes contextual reference to the COVID-19 lockdown measures in the opening sentence (“In response to the adverse impacts of the COVID-19 lockdown measures Move Well, Feel Good (MWFG) was developed as a school intervention using improvement of motor competence as a mechanism for promoting positive mental health”). 

We are grateful for the observations about the structure of the Introduction and the suggestions to amend it. Having reflected on this and discussed the suggestion as an authorship team we feel that the original structure should be retained. In our opinion, it is important to position the contextual references to the COVID-19 lockdown at the beginning of the Introduction, and from there, explain the detrimental impacts on motor competence and mental health which lead into the description of the EESH. We hope that the reviewer can see our perspective on this.

3 As some time has passed since lockdown, I think it would be interesting to include references that support the idea that the effects derived from lockdown still persist. We have included additional references to highlight how the adverse impacts of the lockdowns continued post-pandemic. In lines 110-113 we state “There is a need for intervention strategies in primary school children to address the well-established low levels of motor competence and poor mental health and wellbeing which declined further as immediate and persisting consequences of the COVID-19 lockdown measures (1, 12), even after they ended (6, 20, 21).”

4 The recruitment was deeply explained however I appreciate if the authors could add concrete information about number of participants, schools, mean age, gender, etc. We have now included this information in the opening section of the Results (lines 406-408): “The children were aged 9.6�0.4 years, were predominantly of White British ethnicity (89.8%) with healthy weight status (68.5%), and just over half (51.9%) were girls”.

5 There was a control of the voluntary/optional activities? (The authors stated “which were supplemented by optional class-time, break-time, and home activities”). If yes, please explain it in detail.

 The optional activities were just that and so the research team and teachers had very limited control over how frequently or not the children engaged in these activities. The PE and class teachers highlighted the optional activities to the children and online links to the activity cards were available to the children for home use through QR codes. However, engagement in these activities was completely voluntary.

6 Only three stability skills from the dragon challenge (DC) were used? How do you compute a motor quotient for these three tasks? Please, explain it in detail. 

Why the authors did not consider other alternatives such as KTK? 

In any case, I consider the authors could examine the correlations between CAMSA and DC (3 tasks) to see if both measures are related (as both are supposed to measure participants’ motor competence). The DC includes three stability skills, all of which were included. We therefore were not selective in choosing some skills over others (we presume this is what your reference to “only three stability skills” refers to). As per the DC administration manual the stability scores from these three skills are computed from the sum of the technique and outcome criteria for each skill. The scores can range from 0 to 12. These criteria are detailed in Tyler et al. [2]

The KTK is used to measure gross motor coordination abilities related to dynamic postural balance, which is slightly different to the stability skills that were selected from the DC. The rationale was to supplement the CAMSA with stability skills that are not present within the CAMSA to provide an overall and more holistic measure of motor competence for our study. We acknowledge that the CAMSA includes some dynamic postural balance tasks (i.e., hopping) but does not include the stability skills that are covered within the three DC tasks. In terms of the KTK, two of the DC tasks are similar to two of the subtests from the KTK, however, the KTK does not include core stability/body management, which is encompassed in one of the included DC tasks includes (i.e, core stability). Thus, in order to include a holistic indication of motor competence, and without replicating aspects of motor competence, the CAMSA and DC tasks were chosen.

Further, KTK takes 20–30 min per child to complete all four subtests, which was too long in the context of the available data collection time window we had in each school. Thus, from a feasibility standpoint the KTP was considered but not utilised. The same was true of other motor competence assessments that we considered but were deemed unfeasible due to the anticipated duration taken to complete them.

Theoretically, we could look at the correlation between CAMSA and stability skills from the DC, but the stability skills are there to supplement the CAMSA from a theoretical holistic assessment of motor competence, instead of replacing it. The DC as a whole measures motor competence, but the individual tasks have been shown to individually contribute to assessing aspects of motor competence (see [2]), so theoreticall,y tasks can be used individually to measure specific aspects of motor competence, such as stability. 

To reiterate, we combined the CAMSA and stability skills aspect of the DC to provide a holistic and time-efficient assessment of motor competence that was suitable for this intervention feasibility study.

7 Could you provide further information about the validity and reliability of both tests? I strongly recommend to explain if there was a training period of raters and comparison between them and experts. For the CAMSA, evidence of face validity, convergent validity, and inter-, intra-, and test-retest reliability have been demonstrated and reported by Longmuir et al. [3]. We have now updated the reference citations to include this study (the original reference #36 by the same lead author as the CAMSA study referred to above was mistakenly included). For the DC, evidence for face validity, content validity, and concurrent validity and inter-, intra-, and test-retest reliability have been demonstrated and reported by Tyler et al. [2].

As noted in lines 256-258 the DC assessments and coding were done by trained researchers. Prior to data collection, one of the authors who had substantial experience of administering the CAMSA and DC (RT) trained another author (LC) in the administration and scoring of both assessments. This involved discussing a presentation slide deck outlining the administration and scoring of both assessments, studying the test manuals, and undertaking practice video and live assessments. The training was concluded with LC completing gold standard video assessment trials for both assessments. Agreement with the gold standard scores of > 0.85 were achieved for CAMSA and DC, indicating good-to-excellent reliability.

8 I really would like to know, as a reader, which questions the authors/research team asked to obtain data presented in lines 387-395. Child attendance rate was gathered from class registers taken by the PE teachers. Children’s perceptions of programme enjoyment and teachers’ ratings of programme delivery and child engagement were gathered from the intervention and research methods acceptability surveys detailed in Table 1. These were completed at the end of the programme. 

9 Could you further explain how the focus groups were conducted and recorded? (in which room, if silence was required, etc.) The focus groups took place in a quiet and empty classroom in each school. They were recorded using a digital audio recorder. The children firstly completed the drawing task which focused on their most memorable activities from the project. These drawings were then used as stimuli for the focus group discussions during which the children were encouraged to share their thoughts in an open and safe environment, following question prompts from the researcher. 

10 I really appreciate if the authors could include examples of this result: “All children spoke positively about the motor competence assessments” (line 423). We have now included a focus group quote from one of the children, which reads “I really liked the ones with the jumping in the hoops one-footed, and the balancing spot” (lines 460-461). 

11 Could you provide further statistical results such as mean comparisons (e.g., paired t-test) between T0 and T1? This information could complement the results included in table 4. Further to Reviewer 1’s comments and our responses, we reiterate that the study was not an effectiveness trial and therefore was not powered to assess change in the child-level outcomes. It was for this reason that the pre-post child-level outcomes were presented as descriptive statistics. We did however, present the 95% confidence intervals and where these did not cross zero the results were presented in bold text to indicate ‘meaningful differences’ between the two time points (see Table 4 note). Our approach was guided by the research methodology literature which for feasibility and pilot studies recommends the use of descriptive statistics with confidence intervals rather than formal hypothesis testing with inferential tests and reporting of p values [4, 5].

12 Have the authors check the gender effect? As psychosocial variables are included, I am wondering if changes can be done only in boys or girls. I suggest the authors to consider a model gender*time to see the effects of the MWFG. For the reasons outlined in the response above to comment #11, descriptive analyses were most appropriate for this study. Thus, we did not include a gender*time model. However, if we were to implement a fully powered trial in the future then gender would definitely be included in the statistical analyses.

13 In lines 656-658, the authors stated: “The T1-T0 change scores for overall CAMSA (assessing locomotor and object control skills) (+13.7%) and Dragon Challenge stability skills (+8.1%) and the narrowness of the mean change confidence intervals suggested that the changes were meaningful”. I think that authors should consider my proposal of comments 10 and 11 as they could made assertions based on statistical results. Thank you for this observation. We respectfully refer back to our responses to comments #11 and #12 which also apply here.

14 In lines 659 and 660, the authors mentioned the categories of the CAMSA. Could you provide a supplementary material with descriptive information about how many participants are in each category? We have included this information for boys and girls at both time points in file S6. We have made reference to these additional results and signposted readers to the new Supporting Information File in lines 696-699: “This was reflected in the proportion of children advancing between the CAMSA motor competence categories of ‘Beginning’, Progressing’, Achieving’, and ‘Excelling’ at T0 and T1 (Supporting Information File S6.)”.

Reviewer 3 

1 I am grateful for the opportunity to review this interesting and well written manuscript. The primary purpose of the study is to evaluate the feasibility and acceptability of Move Well, Feel Good (MWFG). Feasibility and acceptability are fundamental to the field of implementation research, and are here examined exemplary well, which leads to results that provide good information about the qualities of the intervention in these areas. 

If possible, I would like to se

---

## [Decision Letter · Decision Letter 1]

18 Apr 2024

Move Well, Feel Good: Feasibility and acceptability of a school-based motor competence intervention to promote positive mental health

PONE-D-23-31624R1

Dear Dr. Fairclough,

We’re pleased to inform you that your manuscript has been judged scientifically suitable for publication and will be formally accepted for publication once it meets all outstanding technical requirements.

Kind regards,

Henri Tilga, PhD

Academic Editor

PLOS ONE

Additional Editor Comments (optional):

Reviewers' comments:

Reviewer's Responses to Questions

**Comments to the Author**

1. If the authors have adequately addressed your comments raised in a previous round of review and you feel that this manuscript is now acceptable for publication, you may indicate that here to bypass the “Comments to the Author” section, enter your conflict of interest statement in the “Confidential to Editor” section, and submit your "Accept" recommendation.

Reviewer #2: All comments have been addressed

2. Is the manuscript technically sound, and do the data support the conclusions?

Reviewer #2: Yes

3. Has the statistical analysis been performed appropriately and rigorously? 

Reviewer #2: Yes

4. Have the authors made all data underlying the findings in their manuscript fully available?

Reviewer #2: Yes

5. Is the manuscript presented in an intelligible fashion and written in standard English?

Reviewer #2: Yes

6. Review Comments to the Author

Reviewer #2: The authors have dealt well with each of my earlier concerns and comments. Overall, the manuscript is clearer now and is relevant for the readership of PLOS ONE. I have no further comment and consider this Manuscript as ready for publication.

7. PLOS authors have the option to publish the peer review history of their article (what does this mean?). If published, this will include your full peer review and any attached files.

Reviewer #2: No

---

## [Editor Report · Acceptance letter]

29 Apr 2024

PONE-D-23-31624R1 

PLOS ONE

Dear Dr. Fairclough, 

I'm pleased to inform you that your manuscript has been deemed suitable for publication in PLOS ONE. Congratulations! Your manuscript is now being handed over to our production team.

Kind regards, 

on behalf of

Dr. Henri Tilga 

Academic Editor

PLOS ONE